

# Bite marks on the frill of a juvenile *Centrosaurus* from the Late Cretaceous Dinosaur Provincial Park Formation, Alberta, Canada

David W.E. Hone[1], Darren H. Tanke[2] and Caleb M. Brown[2]

[1] School of Biological and Chemical Sciences, Queen Mary University of London, London, UK
[2] Royal Tyrrell Museum of Palaeontology, Drumheller, AB, Canada

## ABSTRACT

Bite marks on bones can provide critical information about interactions between carnivores and animals they consumed (or attempted to) in the fossil record. Data from such interactions is somewhat sparse and is hampered by a lack of records in the scientific literature. Here, we present a rare instance of feeding traces on the frill of a juvenile ceratopsian dinosaur from the late Campanian Dinosaur Park Formation of Alberta. It is difficult to determine the likely tracemaker(s) but the strongest candidate is a small-bodied theropod such as a dromaeosaur or juvenile tyrannosaur. This marks the first documented case of carnivore consumption of a juvenile ceratopsid, but may represent scavenging as opposed to predation.

## INTRODUCTION

Bite marks on the bones of fossils can provide important information as to the palaeoecology of ancient ecosystems and as indicators of trophic interactions between animals. In the case of the non-avian dinosaurs (hereafter simply 'dinosaurs'), bite marks (that are healing, healed and peri- or post-mortem) can allow inferences about both inter- and intraspecific interactions in various clades. This includes inferences about cannibalism (*Bell & Currie, 2010*; *Longrich et al., 2010*; *Hone & Tanke, 2015*), scavenging (*Hone & Watabe, 2010*), intraspecific combat (*Tanke & Currie, 1998*), interspecific combat (*Happ, 2008*), prey preferences (*Jacobsen, 1998*), and attempted predation (*De Palma et al., 2013*). However, there are major problems with the use of bite mark data which has limited its potential for interpreting dinosaur behaviour and ecology.

Although tooth-marks are not uncommon for dinosaurs, they are considerably more common in tyrannosaur-dominated faunas (*Fiorillo, 1991*) and can be regularly seen in some formations such as Dinosaur Park Formation (D. Hone, D. Tanke & C. Brown, 2013–2018, personal observation). Even so, relatively few marks have been described in detail to date, which limits comparisons or large-scale assessments of patterns across multiple traces (though see e.g. *Jacobsen, 1998*).

Corresponding author
David W.E. Hone,
dwe_hone@yahoo.com

Identification of both parties associated with bite marks (i.e. both the carnivore and the consumed sensu *Hone & Tanke, 2015*) is often difficult, limiting the available information. Bitten specimens are often fragmentary, and as bite marks are commonly found on isolated elements, these are often not diagnostic to genera or species. Similarly, bite marks are often difficult to attribute to tracemakers (*Hone & Chure, 2018*), although specimens that include shed teeth of a feeding carnivore (*Currie & Jacobsen, 1995*; *Maxwell & Ostrom, 1995*; *Hone et al., 2010*), or where there are single credible candidates for the tracemaker (*Bell & Currie, 2010*; *Longrich et al., 2010*) are known, allowing for a confident referral.

Finally, there are often difficulties in interpreting the actions of the tracemakers based on bite mark data (*Chure, Fiorillo & Jacobsen, 2000*; *Robinson, Jasinski & Sullivan, 2015*). It is difficult to separate out scavenging events from those associated with late stage carcass consumption of a prey item without supporting taphonomic data (*Hone & Watabe, 2010*). Bites may have been made by multiple different tracemaker species, or at different times and traces can potentially be altered through erosion or transport which further restricts interpretations.

Collectively then, this makes interpretations of bite trace data difficult, although it also means that every recorded bite event may be valuable as it is only through the collection and assessment of large datasets that patterns can be assessed. In this context, unusual or rare marks may be especially important for determining the range of possible interactions and events based on theropod bites.

Here, we describe a number of small marks on a partial frill of a juvenile ceratopsian (referred to *Centrosaurus apertus*). Bite marks on ceratopsians are known (*Erickson et al., 1996*; *Jacobsen, 1998*; *Happ, 2008*; *Fowler & Sullivan, 2006*) but are restricted to larger-bodied animals making this the first description of bites on such a young individual. Determining the tracemaker is not possible given the range of possible candidates but this may represent an example of a small-bodied carnivore (i.e. Dromaeosauridae, Troodontidae or juvenile Tyrannosauridae) feeding on the young of a much larger-bodied taxon.

## MATERIALS AND METHODS

The present specimen (Royal Tyrrell Museum of Palaeontology specimen TMP 2014.012.0036) represents a fragment of the squamosal of a subadult centrosaurine ceratopsid (Fig. 1), from the lower Dinosaur Park Formation (Campanian) of southern Alberta. It was found by DHT and collected under Park Research and Collection Permit (No. 14-095) from Alberta Tourism, Parks and Recreation, as well as a Permit to Excavate Palaeontological Resources (No. 14-018) from Alberta Culture and Tourism and the Royal Tyrrell Museum of Palaeontology, both issued to CMB, and is accessioned at the Royal Tyrrell Museum of Palaeontology, Drumheller.

The fossil was collected from the surface of a multi-taxic bonebed in the core area of Dinosaur Provincial Park (UTM, 12U: 464,462 E; 5,621,335 N, WGS 84). Stratigraphically, the specimen is from the lower Dinosaur Park Formation (~5 m above the contact with the underlying Oldman Formation), and falls between the radiometrically dateable Jackson Coulee (min. 76.32 Ma) and Plateau (75.60+/−0.02 Ma) bentonites

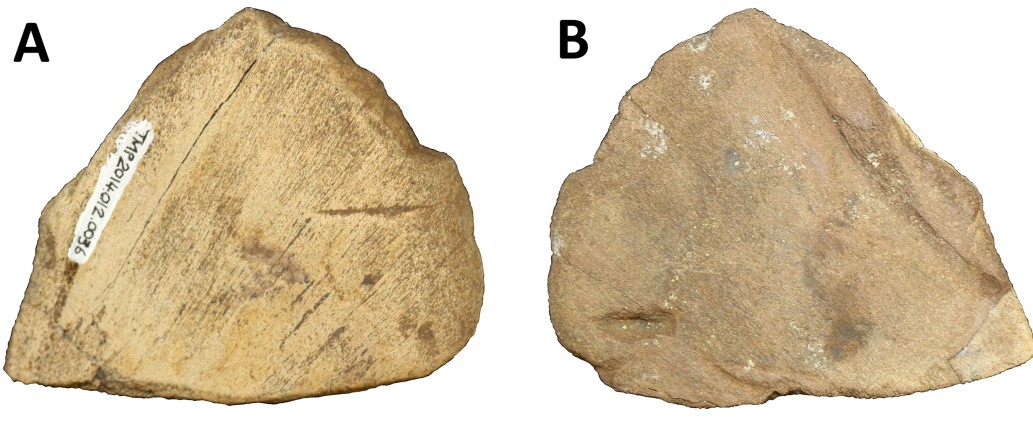

**Figure 1 Photographs of TMP 2014.012.0036 showing side A and side B.** Scale bar is 50 mm long. Image credit: David Hone.

(D. Eberth, 2017, personal communication). This confidently places the specimen within the *Corythosaurus-Centrosaurus* zone (*Ryan et al., 2012*; *Mallon et al., 2013*), and as result, is here referred to *Centrosaurus apertus* as this is the only centrosaurine ceratopsid species known to occur in this well sampled (>20 diagnostic skulls and ~20 bonebeds) interval (*Eberth & Getty, 2005*; *Brown, 2013*).

Multiple systems have been used to describe and define bite marks, and other traces on bones such as trampling, in both the palaeontological and anthropological literature (*Behrensmeyer, Gordon & Yanagi, 1986*; *Hone & Watabe, 2010*). Here, we follow the system of *Hone & Watabe (2010)* as this was created to refer to a series of theropod traces and has been used by a number of different research groups to identify and classify bite marks on dinosaur, and other Mesozoic reptile, bones.

## DESCRIPTION

Specimen TMP 2014.012.0036 is identified as a fragment of squamosal of a small centrosaurine ceratopsid dinosaur (Fig. 1). The specimen is subtriangular in shape and approximately eight cm per side and just over one cm thick. It represents the posterior corner of the lateral margin of the squamosal and is from a position just ventral to the suture with the parietal (Fig. 2). It was broken in several places prior to fossilisation, but part of the original lateral margin remains intact and shows the scalloped edge of the frill.

Four independent lines of evidence suggest this element derived from a non-adult animal. Firstly, despite limited wear to the element, the majority of the surface is unweathered and shows the distinctly striated long grained bone texture of juvenile centrosaurine frill elements (*Sampson, Ryan & Tanke, 1997*; *Brown, Russell & Ryan, 2009*; *Tumarkin-Deratzian, 2010*). Secondly, the preserved lateral margin of the element is straight, and bears no evidence of the imbrication of the loci undulations that develop during ontogeny (*Sampson, Ryan & Tanke, 1997*). Thirdly, the partially preserved epiossification locus is without fused epiossification seen in many (but not ubiquitously

 

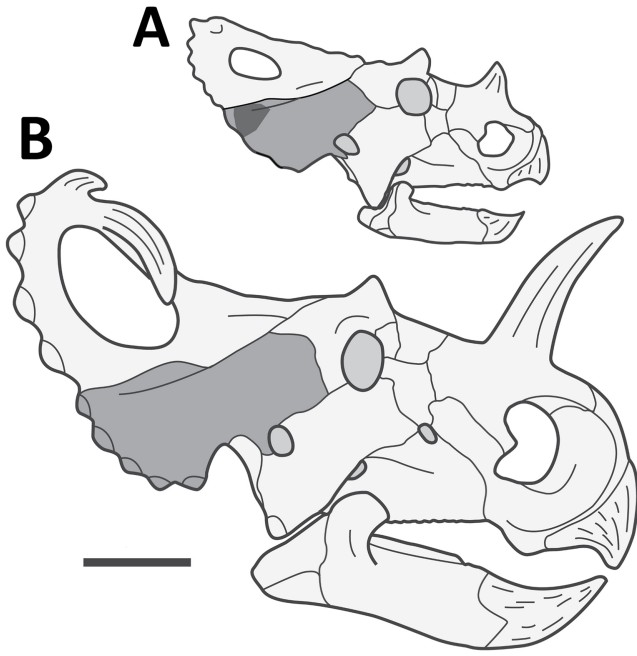

**Figure 2 Reconstructed skull of a juvenile *Centrosaurus apertus*.** Reconstructed skull of a juvenile *Centrosaurus apertus* of approximately similar ontogenetic status to that of TMP 2014.012.0036 (A) in right lateral view, next to that of an adult (B). The two skulls are to scale with one another. The squamosal is highlighted in medium grey and the approximate outline of the specimen preserved here is in dark grey. Scale bar is 200 mm. Reconstruction of the juvenile skull based largely on USNM 7951 (*Gilmore, 1914*), with additions from TMP 1982.016.0011 and 1996.175.0064, adult based on YPM 2015. Image credit: Caleb Brown.

preserved) adults (*Sampson, Ryan & Tanke, 1997*; *Horner & Goodwin, 2008*). Finally, the cross-sectional thickness of the element (<10 mm) and the overall small size of the one preserved episquamosal loci (see Supplementary Data) indicate a small absolute size of the entire squamosal. Taken together, this suggests the animal was below osteologically adult maturity (cf. *Hone, Farke & Wedel, 2016*), and falls into the juvenile age class established by *Sampson, Ryan & Tanke (1997)*.

The absolute size of the animal in life is difficult to estimate from the limited remains, but comparison with a sample of 24 more complete juvenile/subadult squamosals derived from monodominant centrosaurine bonebeds (*Centrosaurus apertus, Coronosaurus brinkmani, Pachyrhinosaurus lakustai*), suggest the complete squamosal would have had a marginal length of approximately 204 mm, and a maximum length of approximately 293 mm. For comparison, osteologically mature *Centrosaurus apertus* specimens have squamosals ranging in marginal length of 258–373 mm (mean = 322 mm), total length of 288–481 mm (mean = 401 mm), for skulls ranging in basal skull length of 660–868 mm (mean = 779 mm). This suggests the tooth-marked squamosal represents an individual with linear skull measures around two-thirds to three-quarters (64–73%) the size of the average ontogenetically adult *Centrosaurus apertus* skull, and approximately one-half (48–61%) the size of the largest *Centrosaurus apertus* skull. Although this may not sound small in comparison, due to the cubic scaling of mass relative linear measures, this equates to an

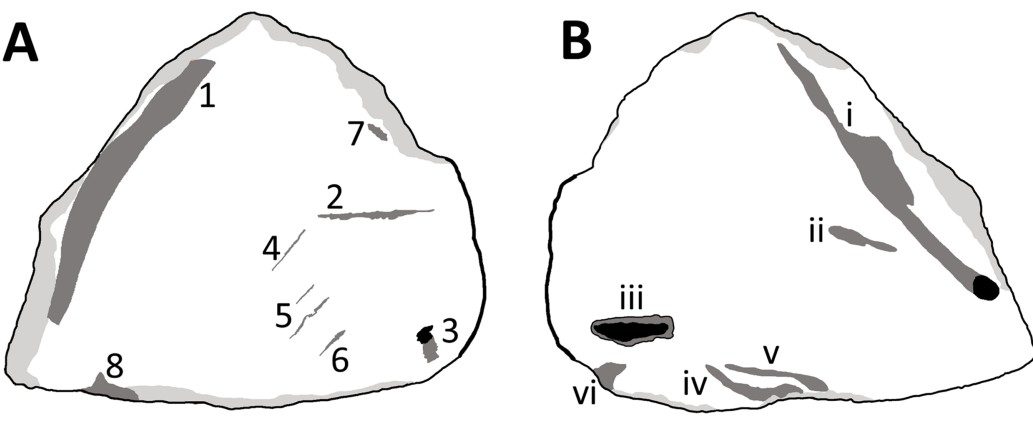

**Figure 3 Interpretative drawing of TMP 2014.012.0036 showing side A (A) and side B (B).** Numbers relate to various areas of interest as described in the text. Pale grey areas mark areas of wear to the bone, dark grey areas represent major features, and black areas are those that penetrate deep into the cortex. The thicker lines on the margins represent the natural margin of the element (see also Fig. 2). Scale bar is 50 mm long. Image credit: David Hone.               

animal less than one-third (~29%), and less than one-seventh (~13%), the mass of the average and largest adult, respectively. This also likely represents an underestimate due to potential negative allometry of the skull relative to the body.

The specimen as preserved has a light coloured and dark coloured side, presumably the former being somewhat bleached by exposure to the sun and rain prior to discovery. The texture on the surface (fine striations) is similar on both sides, suggesting this is a genuine feature and not the result of erosion or exposure. It is not possible to confidently determine which surface is internal and which is external, and as a result, the lighter coloured side is referred to as 'side A,' with the darker side as 'side B' (Fig. 1). A number of features and marks are seen on the specimen that are described below and are numbered as in Fig. 3. Part of the lateral margin of the element is broken (which is common in isolated parts of ceratopsian frills), but one aspect of this retains a natural edge.

Side A (Fig. 3A):

1. A groove on the surface of the bone, which has a counterpart (i) on side B.

2. A thin score that cuts through the cortex. It is long and especially narrow, being 18 by one mm at the widest and mostly circa 0.5 mm wide.

3. A small oval mark (6.5 by three mm) near the margin of the bone. This is uneven and slightly 'Z' shaped.

4–6. A series of marks that resemble cracks. There is some matrix infill of the marks so the margins are not entirely clear. Number 5 is rather irregular and 4 in particular matches other very small cracks in general form.

7. A slight mark on the edge of the bone, near the broken margin. It is small and oval in shape and parallel to the frill margin. The mark is five mm long by 1.5 mm wide.

8. A small but deep mark on the broken margin that is associated with some damage to the frill margin. The mark is five mm long, 1.7 mm deep, and as it is at the broken margin, the width cannot be determined.

Side B (Fig. 3B):

i) A long groove that has some slight damage to one edge of it. This runs parallel to mark 1 on side A.

ii) Two shallow scores, one is broad and the second very thin that departs the former at a shallow angle. The thin side branch does not cut across the fibres of the bone cleanly. The larger trace is 18 mm long and up to 1.25 mm wide.

iii) A short and proportionally deep penetration of the bone, which appears to be broken at the margins. The mark is 11.5 mm long, up to four mm wide and three mm deep (it is deeper proximally and becomes more shallow towards the margin). There is a little wear internally as it is smooth in places including the margins.

iv) A comparatively broad mark that is up to 11.75 mm long, 2.25 mm wide and is approximately one mm deep. The trace is slightly curved along its length.

v) This is a small and narrow score mark that is 17 mm long and one mm wide and closely associated with mark (iv). The depth cannot be measured accurately, but is estimated to be under 0.5 mm. This is subparallel to (ii) and (iii).

vi) A triangular mark that lies at the margin of the piece. The mark is seven mm long, as preserved, and 1.8 mm deep. This lies close to mark (iii).

## DISCUSSION

The specimen here shows a mixture of mark types which are considered to be the result of a combination of effects. The element was found as an isolated piece and not from one of the ceratopsian bonebeds that are common in Dinosaur Provincial Park. Given the isolated nature of the fragment (removed from the rest of the skeleton), and the abraded nature of the breaks, it is likely to have undergone some transport and erosion given that it was not associated with any other parts of a young *Centrosaurus*. This also means that its exact taphonomic history is unknown and thus caution is required when interpreting the limited data.

Breaks to ceratopsian frills are common and thus there is little to take from the separation of the element from the rest of the skull, or the broken margin. Although these are major breaks to this small bone, there is some wear at the edges (suggesting transport and perhaps chemical wear) and the breaks are not clearly associated with possible bites. On side A in particular there are a series of cracks (4–6) on the surface that align with the natural striations on the bone (see Figs. 1 and 3) and the larger manifestations of the long-grained bone texture associated with immature frills (*Sampson, Ryan & Tanke, 1997*; *Brown, Russell & Ryan, 2009*; *Tumarkin-Deratzian, 2010*). Although they are subparallel to each other which is a very common feature of theropod bite marks (*Currie & Jacobsen, 1995*; *Chure, Fiorillo & Jacobsen, 2000*; *Hone & Watabe, 2010*), they also align very well with the general orientation of fibres and smaller cracks on the opposite (B) surface, and are here considered to be aspects of bone growth not alteration. Mark 7 is an odd shape that does not resemble a bite mark and as it is close to the break of the frill margin, it is suggested that this may be part of an impact that lead to

this damage, possibly through trampling (known in some cases to break bones—*Olsen & Shipman, 1988*) or transport. Although different in form, the marks at point (ii) are likely also cracks resulting from the same stress as these also primarily align with the natural form of the bone and the cracks seen on the surface.

Marks 1 and (i) are considered the remains of vascular grooves. They are both broad and shallow and very smooth making them quite unlike typical bite marks. Mark 3 is less clearly defined than others on the bone and the shallow and rounded nature of this make it likely to be part of another vascular groove as with marks 1 and (i).

Marks (ii), (iv) and (v) are difficult to interpret and may be considered bite marks, but this is uncertain. Mark (ii) is slightly tear-drop shaped and does not follow the grain of the bone as with the above marks so it is not part of a crack associated with long grain bone texture. It is however relatively shallow and smooth unlike typical bite marks, although perhaps altered through erosion. This may therefore be the result of a small impact during transport. Similarly, marks (iv) and (v) are subparallel which is a common feature of bite marks. However, they are also rather irregular in shape and do not track each other closely as would be expected for adjacent teeth in a jaw, and mark (iv) has a somewhat sinusoidal pattern. These marks are also smooth and worn, and broad and shallow which is unlike most bite marks, though their identity is unclear. They may be more vascular pathways, or eroded damage, or perhaps both.

Marks 8 and (vi) are relatively deep into the cortex and come at the broken margins of the piece and thus could potentially represent bites that penetrate the cortex and thus may have in part led to the breaking off of the piece. These marks are therefore tentatively assigned as bite marks, but may well be the result of damage from transport and erosion.

This leaves two traces on the specimen that are confidently interpreted as bite marks, trace 2 on the side A and (iii) on side B. Mark 2 is a narrow trace which does correspond in general form to other bite traces seen on bones from the Dinosaur Park Formation (though these are typically considerably larger—D. Hone, 2013–2018, personal observation). This is a long and thin 'diamond' shape tapering to points at each end, although there is also some damage to the margins of this where the bone splintered as the mark was inflicted or perhaps through later erosion. It corresponds to a drag mark (sensu *Hone & Watabe, 2010*) where the tooth does not break through the cortex of the bone. In longitudinal section (Fig. 4) this is deepest in the middle and more shallow at each end and is approximately V-shaped in cross section.

Mark (iii) is close in morphology to a bite and drag (sensu *Hone & Watabe, 2010*) where the tooth penetrates deep into the bone and then is pulled back. This corresponded with the orientation of the bite which is from proximal to distal on the frill being deeper more proximally, and is more shallow towards the frill margin. In cross section this is U-shaped (Fig. 4) and in longitudinal section is seen to be relatively short and deep with the deepest part towards the centre of the element.

## Tracemaker identity

The marks here do not correspond well to those of non-dinosaurian carnivores known from the Dinosaur Park Formation and thus can be ruled out. There are lizards, crocodiles,

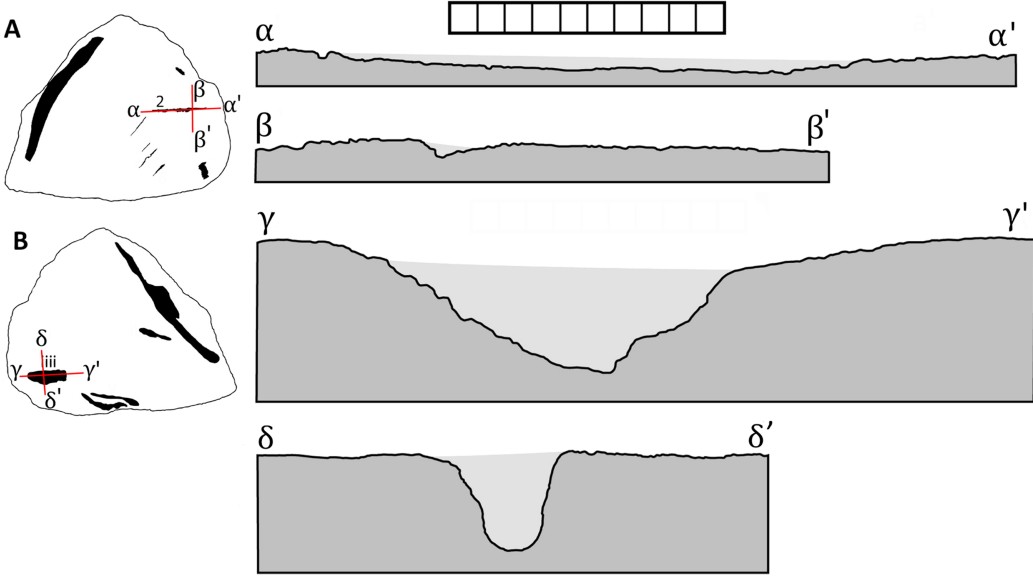

**Figure 4 Interpretative drawings of cross-sections of the traces 2 and iii from TMP 2014.012.0036 based on latex peels showing side A (A) and side B (B).** Dark grey indicates the bone and pale grey the approximate extent of the latex infill. Scale bar is one cm with one mm divisions. Image credit: Caleb Brown.                                               

champsosaurs and mammals known which could potentially have bitten on dinosaur bone. However, extant crocodiles tend to splinter bones when biting and also leave sub-circular punctures not seen here (*Njau & Blumenschine, 2006*; *Drumheller & Brochu, 2014*; *Botfalvai, Prondvai & Ősi, 2014*) and large lizards tend to leave curved traces because the head sweeps in an arc during feeding (*D'Amore & Blumensehine, 2009*). There are no bite marks currently assigned to champsosaurs, but they might be expected to feed in similar ways to either or even both of these techniques (based on their gross anatomy and phylogenetic ancestry) which would not match the traces seen here, and they are widely regarded as piscivorous (*Russell, 1956*). The marks also do not correspond with inferred traces from mammals known from the underlying Oldman Formation of Alberta which appear as repeated pairs of short and wide notches in the bone (*Longrich & Ryan, 2010*).

With these ruled out, the most likely candidates are therefore the non-avian theropods. Three clades of toothed, carnivorous, forms are known from these beds: tyrannosaurs, dromaeosaurs, troodontids as well as the genus *Richardoestesia* which is of uncertain affinities (*Currie, 2005*). Although at adult size, the tyrannosaurs are very large, bite marks from smaller individuals remain a possibility.

Mark 2 is a good match for the very thin and blade-like teeth of dromaeosaurs and troodontids which would leave proportionally thin traces with a narrow V-shaped cross section. Indeed, these marks are a good match in general form for bite marks left by dromaeosaurs in the formation which can be positively identified because of a shed tooth (*Currie & Jacobsen, 1995*). Long and straight bites from tyrannosaurs are typically left as a result of scrape feeding where the premaxillary teeth are drawn across the cortex

(*Hone & Watabe, 2010*) and usually leave multiple subparallel traces that are broad because of the D-shaped nature of the teeth and these are therefore rather unlike mark 2.

The morphology of trace (iii) however, is very different from that of 2, being much more broad and deep and with a U-shaped cross section implying a more blunt tooth made the mark. As noted above, this shape may have been exaggerated by later erosion, but this would still be different to the relatively thin and well-defined trace 2. Although slightly elongate, this is closest to a puncture mark (sensu *Hone & Watabe, 2010*) and would be a good match for a tyrannosaur tooth (premaxillary or maxillary/dentary). Similarly, the traces 3, 8 and (vi), if they are bites, would more closely match tyrannosaurs given their general broad and deep nature. At least some deep puncture wounds that may be attributed to larger dromaeosaurs are known (*Gignac et al., 2010*) and such traces do seem to be relatively rare. Even when a dromaeosaur tooth was punctured into a pterosaur bone with enough force to remove the tooth this was not driven deep into the bone and there were no other associated punctures (*Currie & Jacobsen, 1995*).

The mixture of trace morphology, coupled with the likely erosion of at least some marks makes the identity of the tracemaker difficult to determine. It may have been a dromaeosaurid (cf. *Gignac et al., 2010*) or young tyrannosaur (cf. *Longrich et al., 2010*) or possibly both. Although we are not aware of any bite marks on dinosaur fossils that can be attributed to multiple species this is something which might be predicted—modern carcasses may be fed on by multiple species through kleptoparasitism (*Höner et al., 2002*) or simply feeding on carrion after the original predator has moved on (*Lanszki et al., 2015*).

## Interpretation

In all cases (2, 3, 8, (iii) and (vi)) the traces are well separated from one another and not a series of punctures or sub-parallel marks that are typical of theropod bite traces. Marks may be inconsistent in this regard thanks to the different lengths of theropod teeth in the jaws and possible absences etc. such that a bite may only result in one or two teeth engaging with the bone. In the case of traces 8 and (vi) which abut the broken margins, these may represent a bite on the now missing part of the frill where only a single tooth contacted the squamosal. Single traces made by theropod teeth are certainly known in a number of cases (e.g. some traces in *Erickson & Olson, 1996*; *Tanke & Currie, 1998*; *Gignac et al., 2010*; *Hone & Tanke, 2015*) and so despite the unusual arrangement of these traces, we are confident that several of these do represent bite marks.

Superposition of the two sides of the squamosal piece (Fig. 5) shows that marks 3, (iii) and (vi) are close to one another and 3 and (iii) even partially overlap. However, (iii) lies at a very different angle to the other marks and this is hard to reconcile as being associated with them. In contrast, traces 3 and (vi) are in a similar location and have a similar orientation suggesting they may be the result of a single bite engaging both sides of the frill.

No major muscle groups or abundant soft tissues such as fat deposits are likely associated with the squamosal of ceratopsian dinosaurs. As such, feeding on this part
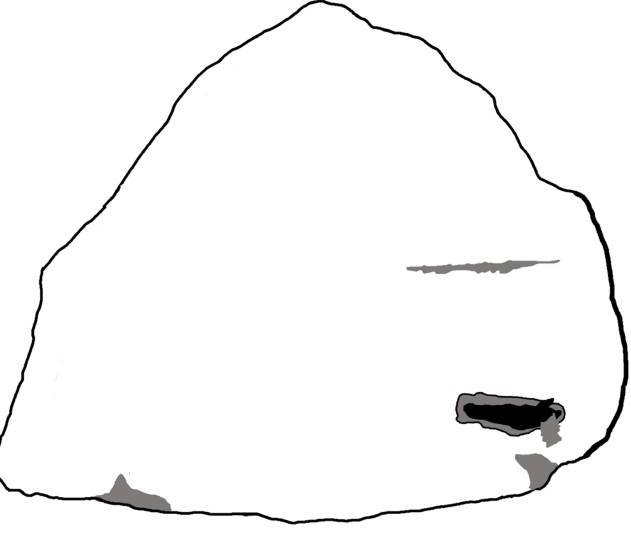

**Figure 5 Interpretative drawing of TMP2014.012.0036 flipped such that the bite marks from the dorsal and ventral sides both appear.** Dark grey areas represent major features, and black areas are those that penetrate deep into the cortex. The thicker line on the margins represent the natural margin of the element (see also Fig. 2). Scale bar is 50 mm long. Photo credit: Caleb Brown.

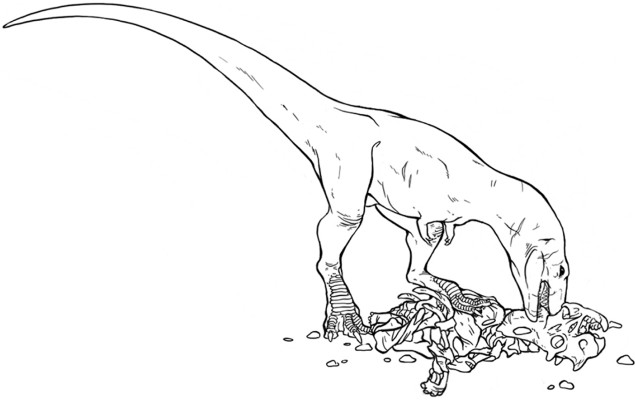

**Figure 6 Speculative reconstruction of scavenging by a juvenile *Gorgosaurus* on the specimen.** Although the identity of the tracemaker of the marks on the *Centrosaurus* frill fragment is uncertain, here we present a speculative reconstruction of scavenging by a juvenile *Gorgosaurus*. Artwork by Marie-Hélène Trudel-Aubry and used with permission.

of the skull was likely a result of late stage carcass consumption (see *Hone & Rauhut, 2010* and references therein) whereby feeding only occurred as a result of the more nutritious aspects of the carcass having been exploited (Fig. 6). The small size of the animal may imply that the carcass was exploited quickly—indeed, large theropods like tyrannosaurs were apparently capable of processing and consuming most or all of a juvenile dinosaur (*Chin et al., 1998*). As a result, although juvenile dinosaurs were likely

common components of dinosaurian faunas, they were at least in part rare in the fossil record as a result of destruction by theropod feeding (*Hone & Rauhut, 2010*). As a result, despite the apparent preferences for feeding on juvenile dinosaurs, most described bite marks are on the bones of adults which may have resisted being consumed and destroyed (even by large tyrannosaurs) and thus feeding traces on a juvenile dinosaur remain unusual. Perhaps the size and shape of ceratopsian crania, even in juveniles, made them difficult to process or required an excess of handling effort for a relatively low reward.

## CONCLUSIONS

Bite marks remain an important source of information on trophic interactions between carnivores and consumed species. Such traces attributed to tyrannosaurs are more common than for other theropod dinosaurs but even so few have been described in detail despite the information that may be available to help interpret their ecology and behaviour. This first evidence of likely scavenging on a non-adult animal adds to the known diversity of animals apparently fed on by Late Cretaceous tyrannosaurs.

## ACKNOWLEDGEMENTS

We thank Marie-Hélène Trudel-Aubry for her artwork as used in Fig. 6. We thank Brandon Strilisky for his help as collections manager and David Eberth for preliminary updated radiometric dates for the specimen. We thank You Hai-Lu, Domenic D'Amore and Stephanie Drumheller-Horton for their comments which improved the manuscript and Mathew Wedel for his handling of this as editor.

### Funding

The authors received no funding for this work.

### Competing Interests

The authors declare that they have no competing interests.

### Author Contributions

- David W.E. Hone conceived and designed the experiments, performed the experiments, analysed the data, prepared figures and/or tables, authored or reviewed drafts of the paper, approved the final draft.
- Darren H. Tanke conceived and designed the experiments, performed the experiments, analysed the data, authored or reviewed drafts of the paper, approved the final draft.
- Caleb M. Brown conceived and designed the experiments, performed the experiments, analysed the data, prepared figures and/or tables, authored or reviewed drafts of the paper, approved the final draft.

## Field Study Permissions

The following information was supplied relating to field study approvals (i.e. approving body and any reference numbers):

Field experiments were approved by a Park Research and Collection Permit (No. 14-095) from Alberta Tourism, Parks and Recreation, as well as a Permit to Excavate Palaeontological Resources (No. 14-018) from Alberta Culture and Tourism and the Royal Tyrrell Museum of Palaeontology.

## Data Availability

The specimen described is stored in Royal Tyrrell Museum of Palaeontology, Drumheller, Alberta, Canada, specimen number TMP 2014.012.0036.

## Supplemental Information

Supplemental information for this article can be found online at http://dx.doi.org/10.7717/peerj.5748#supplemental-information.

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
