# Peer review of "Bite marks on the frill of a juvenile Centrosaurus from the Late Cretaceous Dinosaur Provincial Park Formation, Alberta, Canada"

_PeerJ, doi:10.7717/peerj.5748_

## Round 0.1 · original submission · Minor Revisions

Congratulations - all three reviewers found much of value in your submission, and all three recommended acceptance pending minor revisions. I have read all of the reviewer comments and they are all reasonable and constructive. Please be diligent in addressing them, and I will look forward to seeing an improved version of this work soon.

·

Basic reporting

no comment.

Experimental design

no comment.

Validity of the findings

no comment.

Additional comments

Although bite marks are rare and difficult to interpretate, the authors made great effects to show some of the traces on the frill of a juvenile horned dinosaurs were caused by scavenging of a small-bodied theropod dinosaur based on a good review of previous studies and reasonable explanations. Therefore this is a contribution to our knowledge on dinosaur, esppecially to its paleoecology.
Two suggestions: 1) The discussions on traces ii, iv and v (paragraph beggining at line 201) seem to repeat what just said in previous paragraphs, and these may be combined together. 2) Can you describe what the mammal bite marks would look like as you did for crocs and lizards (paragraph lines 232-243)?
Minor changes can be found in the attached file.

·

Basic reporting

Background information is concise and relevant to the overall thesis, and the structure conforms to PeerJ’s standards. The figures are clear and well labeled with legends that make sense. The authors use clear English. There are several typos within that must be fixed, which I outline below. There is a tendency to use run-on sentences as well, with numerous conjunctions and transitional words. This can be hard to follow at times. Figure 4 could be modified further for clarity. Consider labeling the marks with the same numbers as figure three. Also consider outlining mark 3, so you can more easily distinguish it from mark iii. I really like Figure 5!

Experimental design

The article is original primary research that has not been published before, and will add to the state of the science concerning Mesozoic descriptive taphonomy. The authors clearly have an understanding of the Mesozoic tooth mark literature. I would welcome the inclusion of more anthropological taphonomic literature, but this is not a "dealbreaker." The work falls within the scope of PeerJ, as it is biological in nature and a research article. The research question is well defined, in that it is a descriptive study of taphonomic traces with the attempt to describe them and identify the potential tracemarker(s). The methods are replicable for the most part, but the way they determine that the fossil was from a juvenile Centrosaurus could use more elaboration. The authors state they compared it with other centrosaurines and provide some data ranges, but they do not say which specimens. The authors should add a supplemental table of the specimens sampled, or at very least state the collections or dig sites these specimens were taken from.

Validity of the findings

Data is robust. The authors draw conclusions about the traces that are reasonable and well supported, and are very clear about what is speculation.

Additional comments

Abstract:
• Line 11: this sentence is awkward with the position of the parenthetical statement. Rephrase.
• Line 13: the use of “and” seems inappropriate. What is “hampered” exactly? The ability to produce robust conclusions from traces? Adequate sample?
• Line 16: Fix typo “a s dromaeosaur”
Introduction
• Line 23: missing an apostrophe after “dinosaur”. Consider rephrasing the parenthetical statement to “including those that are healing, healed, and per- or post-mortem”
• Line 29: “potential” to do what?
• Line 33: change to “…described in detail”
• Line 43: change to “…difficulties in interpreting the actions of the tracemarkers based on bite mark data”
• Line 47: A bite is rarely made by multiple tracemarkers. Specify you are talking about multiple bites on a single bone/carcass/assemblage.
• Line 49: Run-on sentence
Materials and Methods
• Line 64: Run-on sentence. Consider breaking before “both”
• Line 75: change to “…and, as a result, is”. Eliminate the second “this is”
Description
• Line 85: change to “…from a juvenile /subadult animal.”
• Line 89: “in” should be “is”
• Line 91: No comma after “loci”
• Line 95: change to “they suggest” or “this suggests”
• Paragraph beginning at line 98: see comments on fossil specimens in ‘Experimental Design” section.
• Line 111: Last sentence of the paragraph does not make sense. Rephrase
• Line 126: what is “c.”?
• Line 127: mention if this mark fully penetrates cortical bone here (you say it in other places).
• Line 135: put a comma after “and”
• Line 143: similar to above, state if this fully penetrated the cortex.
Discussion
• Line 156: eliminate “of”
• Line 158: I think there should be a parenthesis after “skeleton”. Consider breaking up this sentence
• Line 169: parenthesis after “marks”.
• Line 171: you state you don’t know which side of the fossil is which, then how can you justify saying it is “dorsal”?
• Line 173: there is some lit on trample marks you may want to consult before you conclude trampling. Olsen and Shipman (1988: J. Archeological Science) wrote one of the first papers on the subject.
• Line 179: Run-on sentence.
• Line 181: Fix indent.
• Line 187: put “it” after “so”
• Line 188: should “and” be there?
• Line 194: Confusing sentence. Rewrite.
• Line 207: You can either say “proximal to distal” or “medial to lateral.” I think you mean the former.
• Line 209: Do you mean expanded closer to the margin of the frill? Can you use it as an anatomical reference point?
• Line 213: change “no” to “not”
• Line 219: Run-on sentence.
• Line 222: reference the idea they were piscivorous
• Line 233: Run-on sentence.
Conclusion
• Line 288: “traces attributes” is confusing
References:
• Line 323: italicize Alligator mississippiensis

·

Basic reporting

“Bite marks on the frill of a juvenile Centrosaurus from the Late Cretaceous Dinosaur Provincial Park Formation” describes a set of different bone surface modifications on this skeletal element, with descriptions and interpretations of each. In general, the paper is well-organized and written (see my very minor comments listed below regarding spelling, punctuation, etc.). All information needed for replicating the results is included in the paper.

The introduction and background are both generally well-written, but seem to be missing some highly relevant literature. The authors do cite D’Amore and Blumenschine, 2009, but while discussing the scarcity of papers describing theropod bite marks in the fossil record, fail to cite many of the papers listed in Table 1 of the D’Amore and Blumenschine study (a fairly exhaustive list of theropod bite mark papers, as of that year). Other relevant papers missing from the background and later discussion are Longrich et al., 2010 (not the same one which is already in the text), which covers an incidence of scavenging by a juvenile T. rex (a topic which is important to the findings of this paper), and Gignac et al., 2017, which itself describes and interprets T. rex bite marks, but also cites several other case studies and surveys of theropod bite marks which should be included in this manuscript. There was also an abstract by Fowler and colleagues in 2012, which addresses patterns of T. rex bite marks on several different Triceratops specimens, especially skulls. While this is gray literature, the authors might consider reading and citing it as well, because it specifically discusses frill-biting behavior, which is of special interest to the study at hand. In short, theropod bite marks actually are better known and better documented than this manuscript currently indicates, and this growing body of literature is allowing not just documentation of specific trophic events, but discussions of larger-scale patterns across clades. Including discussions of these other papers will not reduce the novelty of the presented findings, but it will provide more context for these bigger picture questions related to bite marks and feeding behavior/trophic structure.

Perhaps a larger problem is the lack of any citations for the extensive, relevant anthropological literature on bite marks and other bone surface modifications. This is a fairly common issue in paleontological papers, which has resulted in our community reinventing the wheel with regards to many bone surface modification studies. I will discuss other specifics in more detail below, but in the context of the general framing and background of the paper, explicit guidelines for identifying and classifying bite marks relative to other bone surface modifications were erected by Binford in 1981, and their repeatability tested using blind inter-analyst studies of correspondence by Blumenschine et al., in 1996. Binford’s classification scheme has been widely used in anthropology ever since, and it is starting to be widely adopted in paleontology as well (see Njau and Blumenschine, 2006, D’Amore and Blumenschine, 2009, and Drumheller and Brochu, 2014 for examples, all papers which were already cited in this manuscript). These classifications predate the ones used in this study, but more problematic is that the authors use some of the same terminology (e.g. “score,” “furrow”) used in Binford’s scheme, but using different definitions. Being explicit with the classification scheme up front and avoiding these terms with conflicting or imprecise definitions (or simply using the Binford scheme) would increase the clarity of the background and descriptive sections of this paper.

The topics addressed in the paragraph starting on line 43 would also benefit from including discussions of previous studies pulled from the anthropological or non-dinosaurian, paleontological literature. There are several published examples of multiple feeding trace makers interacting with a single prey item, with discussions on how to differentiate them and how to determine order of access to remains (e.g. Brain, 1980; Haynes, 1983; Selvaggio, 1998; Selvaggio and Wilder, 2001; Dominguez-Rodrigo and Piqueras, 2003; Delaney-Rivera et al., 2009; Drumheller et al., 2014). The authors mention that erosion or transport might be obscuring bone surface modifications, but do not cite examples of studies exploring the effects these processes have on interpreting surficial traces (e.g. Behrensmeyer, 1978; Behrensmeyer and Boaz, 1980; Behrensmeyer, 1981; Fisher, 1995). Difficulties in identifying scavenging are mentioned in the same paragraph, a discussion which could be strengthened by reading and citing some of the following papers: Hill, 1976; Shipman and Phillips, 1976; Shipman, 1986; Dominguez-Rodrigo, 1999; Forrest, 2003; Longrich et al., 2010. The addition of some or all of these studies would strengthen many of the subsequent interpretations presented in this paper.

While I generally like the figures, I am a little concerned with the main description of Figures 3 and 4 being found only in the text, but not in the figure captions themselves. I had to flip back and forth between the description and the figures, which wasn’t ideal for ease of reading and comprehension. I understand that including all of that material in a figure caption would make it very long, and perhaps this issue will be lessened when the manuscript is formatted for publication, which will bring the descriptions closer to the figures themselves. Also, I am unsure what Figure 4 adds that could not be included (or isn’t already included) in Figure 3B. Lastly, some of the major bone surface modifications are visible in Figure 1, but others are not (or are at least difficult to see). Would it be possible to include higher magnification images for clarification in either Figure 1 or 3?

Minor issues with punctuation, grammar, etc.
Line 16 – “as” has a space between the two letters
Line 62 – the authors actually use the repository abbreviation before defining it
Line 93 – extra “the” between “cross-sectional” and “thickness”
Line 111 – “and understatement” should be “an understatement”
Line 131 – delete indentation
Line 185 – comma between “marks” and “but”
Line 289 – comma between “dinosaurs” and “but”
Line 323 – italicize “Alligator mississippiensis”
Line 359 – hyphen seems to have been replaced with an error symbol
Line 375 – “Njau” is misspelled
Line 379 – check “?;” punctuation in article title

Experimental design

The fossil described in this paper is properly reposited, and the authors cite all relevant permits, demonstrating that the specimen was obtained legally and ethically. The authors provide very specific location information for where the specimen was recovered. I know that some authors (and repositories) prefer to keep these kinds of details out of papers, so as not to tip of fossil poachers. However, two of the authors work for the museum in question, so I am sure they are more familiar with their institution’s feelings on such matters.

The analysis itself is a descriptive treatment of the fossil specimen with comparisons to other, similarly marked specimens in the literature. Other than the terminology and background issues I covered above, I think that this aspect of the paper is well done and certainly in keeping with other examples of similar studies in the literature.

Validity of the findings

Overall, the findings of this paper are largely supported by the data, as presented. The authors are conservative with their interpretations of the bone surface modifications they see, and do not go beyond their dataset. What issues I do have with this section relate to terminology (see the discussion of the Binford classification scheme, etc. above) and a need for more detail in places.

The bone surface modifications on Side A, 4-6 are said to “resemble cracks” (line 129, and the paragraph starting on line 162). More detail here would be nice. Do they exhibit any crushing damage, which is an essential part of diagnosing bite marks sensu Binford, 1981 and Blumenschine et al., 1996? It sounds like they do not, and are more related to cracking along the fabric of the bone, as described in papers like Behrensmeyer, 1978, which discusses the details of taphonomic history which can be gleaned from weathering patterns. As the bone surface modifications which are not identified as bite marks are not discussed in much detail later, a few lines of discussion here would really add to the overall description of the fossil.

The authors mention trampling as a potential source of mark A7 in the same paragraph, but do not cite or compare the features to Behrensmeyer et al., 1986, which describes trampling damage to bone in detail. Again, adding a brief discussion and a citation would be helpful. The paragraphs that starts on lines 185 and 194 has a similar lack of detail or discussion. Do the marks described exhibit signs of crushing or other impact damage? If not, they don’t meet the criteria for identification as a bite mark. Could they be other types of bone surface modifications instead (see Fisher, 1995 for several examples)? None of this is really discussed.

The structures which have been more confidently identified as bite marks, starting on line 198, could also use a bit more detail in their descriptions. The drag mark (line 203) corresponds to a score, as defined by Binford, but detail beyond that could help bolster their association with a specific actor later on in the paper. What shape are they in cross section? U-shaped bite marks are characteristic of most bite marks, but add a bisection, and you have the defining characteristics of traces left by the highly carinated teeth of crocodylians (Njau and Blumenschine, 2006; Drumheller and Brochu, 2014; Drumheller and Brochu, 2016). More v-shaped traces might be related to the laterally compressed teeth of dinosaurs, especially if striations (associated with tooth serrations) are present (D’Amore and Blumenschine, 2009; 2012). Striations are not mentioned at all in the description. If they are absent, that is worth noting, but does not necessarily mean that the marks weren’t left by a typical theropod tooth. If they are present, then the authors might be able to better refine their association with an actor, following the methods presented in D’Amore and Blumenschine, 2009 and 2012. Striations and bisections should also be mentioned, if only to note their absence, in the section identifying the trace makers, starting on line 212. More details discounting mammals are also warranted too. On line 222, the authors say that none of the marks correspond with known bite marks attributable to Cretaceous mammals, but provide no further discussion or detail.

Mark Biii certainly needs a more detailed description. The authors themselves say they are “not sure this is explained well here – hard to describe” on line 209, a note I am not sure they meant to leave in the submitted text. I believe that I understand what they are saying though. The surface margins of the feature are inset in comparison to the wider, underlying structure. Again, especially in the absence of crushing damage, this doesn’t sound like it necessarily is a bite mark. The sediment scouring they suggest as an explanation could widen and deepen any number of other bone surface modifications or damage. If there is crushing damage, adding a description of it would bolster their current interpretation. The fact that there is a similar mark on the opposite site is also potential support for their interpretation. I am not necessarily disagreeing with their interpretation, just saying that I think it needs a little more detail to be convincing. As mentioned in the figure discussion above, these features are hard to see and judge in the current Figure 1, so a higher magnification image of the most important of surficial structures might help.

The paragraph starting on line 249 mentions the possibility of multiple actors leaving bite marks on a single bone. Please reference my suggestions above regarding similar studies that could be cited to bolster the arguments in this paragraph. The paragraph starting on line 272 addresses order of access to remains, or order of remains consumption. Please reference my section discussing scavenging above, and also add to that papers on scavenging/predation sequences, such as Hill, 1980, Binford, 1981, Haglund, 1997, and Behrensmeyer et al., 2003. The paragraph starting on line 272 might also benefit from a comparison to the Fowler et al study mentioned above, but again, that is from the gray literature, and I understand if the authors want to avoid citing it.

Additional comments

In general, this paper is well organized and conceived, but it is a little underdeveloped and could be strengthened considerably with the addition of more descriptive details, especially following (and citing) the findings of several of the additional papers listed in the previous sections. This will allow interpretations that are currently presented as speculative to be reframed with more context and support. With these additions, I think that this paper will be a strong addition to the growing body of literature describing and interpreting theropod bite marks.

Citations not already included in the manuscript:

Behrensmeyer, A.K., 1978. Taphonomic and ecologic information from bone weathering. Paleobiology, 4: 150-162.

Behrensmeyer, A.K., 1981. Vertebrate paleoecology in a Recent East African ecosystem. In Communities of the Past. Eds. Gray, J., A.J. Boucot, and W.B.N. Berry. Stroudsburg, PA: Dowden, Hutchinson & Ross, 591-615.

Behrensmeyer, A.K. and D.E.D. Boaz, 1980. The Recent bones of Amboseli National Park, Kenya, in relation to East African paleoecology. In Fossils in the Making: Vertebrate Taphonomy and Paleoecology. Eds. Behrensmeyer, A.K. and A.P Hill. Chicago: The University of Chicago Press, 72-93.

Behrensmeyer, A.K., K.D. Gordon, and G.T. Yanagi, 1986. Trampling as a cause of bone surface damage and pseudo-cutmarks. Nature, 319: 768-771.

Behrensmeyer, A.K., C.T. Stayton, and R.E. Chapman, 2003. Taphonomy and ecology of modern avifaunal remains from Amboseli Park, Kenya. Paleobiology, 29(1): 52-70.

Binford, L.R., 1981. Bones: Ancient Men and Modern Myths. New York: Academic Press.

Blumenschine, R.J., C.W. Marean, and S.D. Capaldo, 1996. Blind test of inter-analyst correspondence and accuracy in the identification of cut marks, percussion marks, and carnivore tooth marks on bone surfaces. Journal of Archaeological Science, 23(4): 493-507.

Brain, C.K., 1980. Some criteria for the recognition of bone-collecting agencies in African caves. In Fossils in the Making: Vertebrate Taphonomy and Paleoecology. Eds. Behrensmeyer, A.K. and A.P Hill. Chicago: The University of Chicago Press, 108-130.

D’Amore D.C. and R.J. Blumenschine, 2012. Using striated tooth marks on bone to predict body size in theropod dinosaurs: a model based on feeding observations of Varanus komodoensis, the Komodo monitor. Paleobiology, 38(1): 79-100.

Delaney-Rivera, C., T.W. Plummer, J.A. Hodgson, F. Forrest, F. Hertel, and J.S. Oliver, 2009. Pits and pitfalls: taxonomic variability and patterning in tooth mark dimensions. Journal of Archaeological Science, 36: 2597-2608.

Domínguez-Rodrigo, M., 1999. Flesh availability and bone modifications in carcasses consumed by lions: paleoecological relevance in hominid foraging patterns. Palaeogeography, Palaeoclimatology, and Palaeoecology, 149: 373-388.

Domínguez-Rodrigo, M. and A. Piqueras, 2003. The use of tooth pits to identify carnivore taxa in tooth-marked archaeofaunas and their relevance to reconstruct hominid carcass processing behaviors. Journal of Archaeological Sciences, 30: 1385-1391.

Drumheller, S.K. and C.A. Brochu, 2016. Phylogenetic taphonomy: A statistical and phylogenetic approach for exploring taphonomic patterns in the fossil record using crocodylians. Palaios, 31(10: 463-478.

Drumheller, S.K., M.R. Stocker, and S.J. Nesbitt, 2014. Direct evidence of trophic interactions among apex predators in the Late Triassic of North America. Naturwissenschaften, 101(11): 975-987.

Fisher, J.W., 1995. Bone surface modifications in zooarchaeology. Journal of Archaeological Method and Theory, 2(1): 7-68.

Forrest, R., 2003. Evidence for scavenging by the marine crocodile Metriorhynchus on the carcass of a plesiosaur. Proceedings of the Geologists’ Association, 144: 363-366.

Fowler, D.W., J.B. Scannella, M.B. Goodwin, and J.R. Horner, 2012. How to eat a Triceratops: large sample of toothmarks provides new insight into the feeding behavior of Tyrannosaurus. Journal of Vertebrate Paleontology, 32: 96.

Gignac, P.M. and G.M. Erickson, 2017. The biomechanics behind extreme osteophagy in Tyrannosaurus rex. Scientific Reports, 7: doi:10.1038/s41598-017-02161-w.

Haglund, W.D., 1997. Dogs and Coyotes: Postmortem Involvement with Human Remains. In Forensic Taphonomy: The Postmortem Fate of Human Remains. Eds. W.D. Haglund and M.H. Sorg. Boca Raton: CRC Press, 367-381.

Haynes, G., 1983. A guide for differentiating mammalian carnivore taxa responsible for gnaw damage to herbivore long bones. Paleobiology, 9(2): 164-172.

Hill, A., 1976. On carnivore and weathering damage to bone. Current Archaeology, 17(2): 335-336.

Longrich, N.R., J.R. Horner, G.M. Erickson, and P.J. Currie, 2010. Cannibalism in Tyrannosaurus rex. PLoS ONE, 5(10): e13419.

Selvaggio, M.M., 1998. Evidence for a three-stage sequence of hominid and carnivore involvement with long bones at FLK Zinjanthropus, Olduvai Gorge, Tanzania. Journal of Archaeological Science, 25: 191-202.

Selvaggio, M.M. and J. Wilder, 2001. Identifying the involvement of multiple carnivore taxa with archaeological bone assemblages. Journal of Archaeological Science, 28: 465-470.

Shipman, P., 1986. Scavenging or hunting in early hominids: theoretical framework and tests. American Anthropologist, New Series, 88(1): 27-43.

Shipman, P. and J.E. Phillips, 1976. On scavenging by hominids and other carnivores. Current Anthropology, 17(1): 170-172.

---

## Round 0.2 · Minor Revisions

Thank you for addressing in the revised manuscript and rebuttal letter all of the concerns raised by the reviewers. In general I am satisfied with your responses to the reviewers, so I don't think another round of external review is warranted. Nevertheless, there are a few issues yet to deal with.

The issue of classifications used for bite marks requires more attention. In your rebuttal letter you wrote, "We use the system outlined by Hone & Watabe ... We make it quite clear which naming system we are using and cited this". Although you cite Hone & Watabe, if you came right out and stated explicitly that you were using that system, I missed it. Since this is a potentially confusing area for readers who are not intimately acquainted with the literature, it needs to be addressed clearly. Please add a section to the Materials and Methods in which you minimally (1) acknowledge that more than one set of terminology is "in play" in the historical sciences, (2) say explicitly which one you are using, and (3) justify that decision. If the anthropological one is inappropriate, given a compelling reason why. I don't think you are wrong here, I just want you to make it clear for readers which system you're using and why.

The revised manuscript also has many sentences that are awkwardly worded, or that need to be broken up into separate clauses by commas, or in which pronouns have no clear antecedents. A few examples (lines refer to the reviewing PDF, not the Word doc):
- the sentence that runs from lines 29-32 mentions cannibalism twice - was that intentional?
- The sentence in lines 43-47 is particularly difficult to follow, especially the concluding words "support identifications" - does that refer to all previous clauses, or only the last? Please reword for clarity.
- line 68 - no space between specimen and TMP
- the sentence starting in line 88 needs a subject - as is, "This" would most naturally refer to "a small centrosaurine ceratopsid dinosaur" from the end of the previous sentence, rather than the specimen (which I think you're referring to) from the start. Better to just start the sentence with "This specimen" or "The specimen".

This list is illustrative, not exhaustive. Please read the entire manuscript carefully (out loud, if necessary) to clean up the prose.

Finally, two minor changes to the figures would improve their readability.
- Figure 1 would be more useful if the orientation was clarified. Please label the external margin, the contact with the parietal (if there is one - Figure 2 suggests there is), and the third edge with whatever the appropriate anatomical term might be (i.e., dorsal/ventral, medial/lateral, rostral/caudal).
- The length of the scale bar in Figure 2 is not given.

Once these areas are addressed, I see no barrier to the acceptance of your manuscript for publication in PeerJ.

---

## Round 0.3 · accepted · Accept

Thank you for your diligence in addressing the last round of edits. I am happy to accept your manuscript for publication in PeerJ. (One thing to note when you upload the final version: in the caption for Figure 2, "Scale bar" is misspelled as "Scale bare".)

The decision of whether or not to publish the peer reviews alongside the paper is entirely yours, and will not affect how your paper is handled going forward. However, I encourage you to do so. Making the reviews public allows the reviewers to receive more credit for their efforts, and also contributes to the emerging culture of fairness and transparency in editing and peer review.